# Immunotherapy with Antibodies in Multiple Myeloma: Monoclonals, Bispecifics, and Immunoconjugates

Christie P. M. Verkleij [ID], Wassilis S. C. Bruins, Sonja Zweegman and Niels W. C. J. van de Donk *[ID]

Department of Hematology, Cancer Center Amsterdam, Amsterdam UMC, Vrije Universiteit Amsterdam, 1081HV Amsterdam, The Netherlands; c.verkleij@amsterdamumc.nl (C.P.M.V.); w.s.bruins@amsterdamumc.nl (W.S.C.B.); s.zweegman@vumc.nl (S.Z.)
* Correspondence: n.vandedonk@amsterdamumc.nl; Tel.: +31-(0)-20-4442604

**Abstract:** In the 2010s, immunotherapy revolutionized the treatment landscape of multiple myeloma. CD38-targeting antibodies were initially applied as monotherapy in end-stage patients, but are now also approved by EMA/FDA in combination with standards-of-care in newly diagnosed disease or in patients with early relapse. The approved SLAMF7-targeting antibody can also be successfully combined with lenalidomide or pomalidomide in relapsed/refractory myeloma. Although this has resulted in improved clinical outcomes, there remains a high unmet need in patients who become refractory to immunomodulatory drugs, proteasome inhibitors and CD38-targeting antibodies. Several new antibody formats, such as antibody–drug conjugates (e.g., belantamab mafodotin, which was approved in 2020 and targets BCMA) and T cell redirecting bispecific antibodies (e.g., teclistamab, talquetamab, cevostamab, AMG-420, and CC-93269) are active in these triple-class refractory patients. Based on their promising efficacy, it is expected that these new antibody formats will also be combined with other agents in earlier disease settings.

**Keywords:** multiple myeloma; immunotherapy; antibodies; monoclonal; bispecific; immunoconjugates; antibody-drug conjugates





## 1. Introduction

The survival of multiple myeloma (MM) patients has substantially improved over the last three decades because of the introduction of autologous stem cell transplantation and novel agents, such as immunomodulatory drugs (IMiDs; e.g., thalidomide, lenalidomide, and pomalidomide) and proteasome inhibitors (PIs; e.g., bortezomib, ixazomib, and carfilzomib). More recently, the incorporation of CD38- and SLAMF7-specific antibodies in treatment regimens for patients with newly diagnosed or relapsed/refractory disease, has further improved the clinical outcomes of MM patients. Based on the activity and favorable toxicity profile of these naked antibodies, several new antibody formats are now evaluated in clinical trials in extensively pretreated patients. In this review, we will discuss the efficacy and safety profile of naked antibodies as well as novel antibody formats such as bispecific antibodies and immunoconjugates.

## 2. Naked Antibodies

### 2.1. CD38-Targeting Antibodies

Naked CD38-targeting antibodies (approved: daratumumab and isatuximab; in clinical development: MOR202, SAR442085, and TAK-079) induce MM cell death via direct on-tumor effects such as the direct induction of apoptosis, complement-mediated cytotoxicity (CDC), antibody-dependent cellular cytotoxicity (ADCC), and antibody-dependent cellular phagocytosis (ADCP) (Figure 1) [1–3]. Although there is overlap in the mode of action of these antibodies, there are also some differences [3]. Daratumumab is most potent in terms of the induction of CDC, while isatuximab is more effective in the direct induction of cell death [3]. In addition, CD38-targeting antibodies have T cell stimulatory properties

by eliminating CD38-positive regulatory T cells, regulatory B cells, and myeloid-derived suppressor cells [4–7].

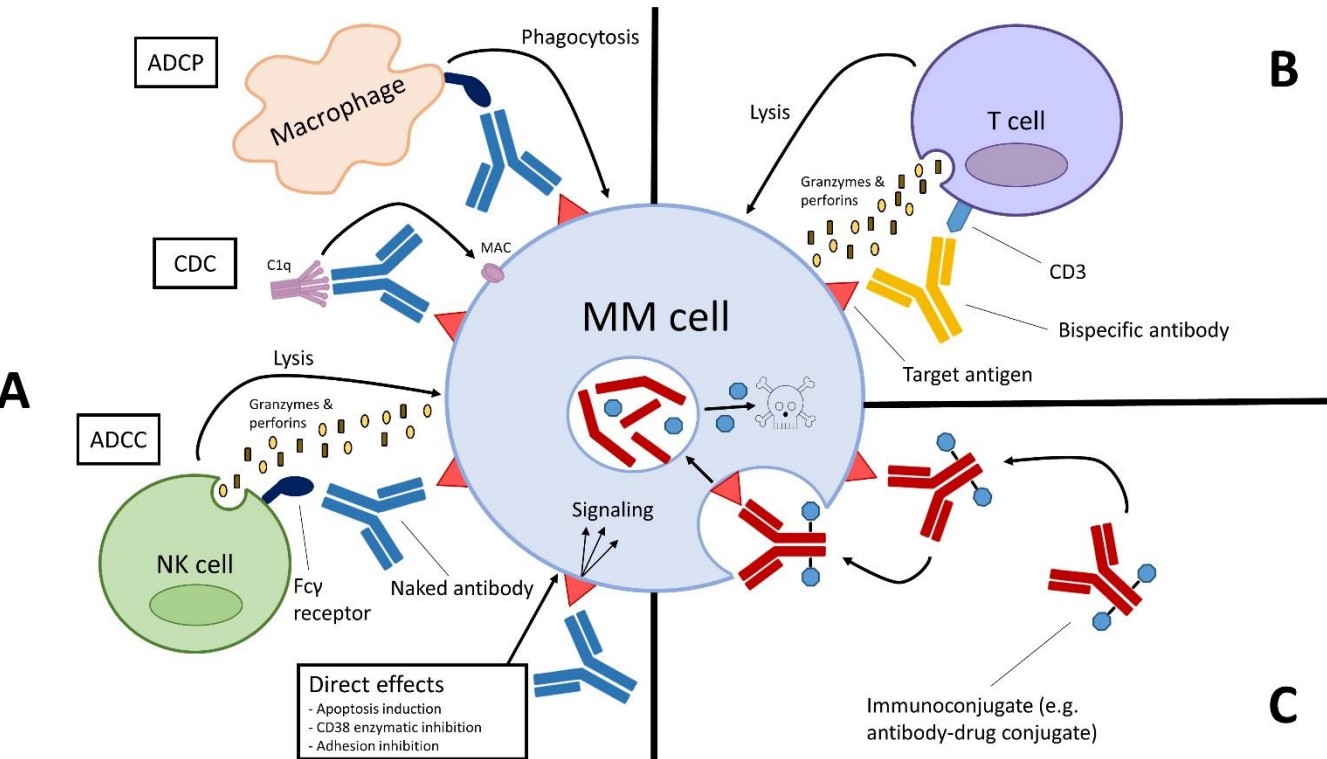

**Figure 1.** Immunotherapy with antibodies in multiple myeloma. Mode of action of naked antibodies (**A**), bispecific antibodies (**B**), and immunoconjugates (**C**). Abbreviations: MM, multiple myeloma; CDC, complement-mediated cytotoxicity; ADCC, antibody-dependent cellular cytotoxicity; ADCP, antibody-dependent cellular phagocytosis; MAC, membrane attack complex.

CD38-targeting antibodies were first explored as monotherapy in patients with disease exposed to IMiDs and PIs. At that time, these double-refractory patients had a very poor outcome with a median overall survival of only nine months [8]. The overall response rate with CD38-targeting antibodies as a single agent was approximately 30% [9–12]. Importantly, long-term follow-up of the GEN501 and Sirius studies, which evaluated daratumumab monotherapy in patients with advanced MM, showed a remarkably long overall survival (median overall survival: 20.5 months, with a three-year overall survival rate of 36.5%) [13]. This survival is much longer than what was observed in a similar patient population, who did not receive CD38-targeted therapy [8]. Equally important is the favorable toxicity profile of CD38-targeting antibodies. The most frequent side effect observed with CD38-targeting antibodies as monotherapy is the development of infusion-related reactions, most often observed during the first infusion. Premedication with acetaminophen, antihistamine and a steroid is important to prevent infusion reactions, which are characterized by fever, chills, coughing, and sometimes dyspnea. The leukotriene receptor antagonist montelukast is also helpful to prevent infusion-related reactions. Another issue of CD38-targeting antibodies is that these therapeutic antibodies can be detected in laboratory assays such as serum protein electrophoresis (SPEP) and immune fixation electrophoresis (IFE) assays. When the therapeutic antibodies co-migrate with the M-protein of the patient, and are of the same isotype, laboratory personnel may be unable to differentiate between a very good partial response and a complete response. The daratumumab interference reflex assay (DIRA) is able to shift the migration pattern of daratumumab, which enables the correct quantification of the patient's M-protein [14–17]. Blood banks also need to be informed when a patient is treated with a CD38-targeting antibody. Red blood cells express low levels of CD38; thus, CD38-targeting antibodies

interfere with the indirect Coombs test, which is used by blood transfusion laboratories to assess the presence of anti-red blood cell antibodies [18]. Several mitigation strategies are now available to solve this issue, including the phenotyping (before start of CD38-targeting antibody treatment) or genotyping of clinically relevant red blood cell antigens [16]. Furthermore, the use of dithiothreitol (DTT) to remove CD38 on the surface of red blood cells can be used to safely provide blood to CD38-targeting antibody-treated patients. Because DTT denatures Kell antigens, K-negative units are provided to these patients [18].

### 2.2. Combination Therapy with CD38-Targeting Antibodies

Based on the excellent balance between safety and activity, CD38-targeting antibodies are attractive partners for combination strategies in both newly diagnosed and relapsed/refractory patients.

### 2.2.1. IMiD-Based Combinations

The MAIA study [19], which enrolled newly diagnosed, transplant ineligible MM patients, and the POLLUX [20] study, which included patients with at least one prior line of therapy, showed that adding daratumumab to the standard-of-care regimen lenalidomide–dexamethasone (Rd) improved response rates, including the proportion of patients with minimal-residual disease negative complete remissions (Table 1). The superior response rate translated into a longer progression-free survival of the daratumumab-based triplet, compared to Rd alone. Although cross-trial comparisons should be performed with caution, given the heterogeneity in patient populations, in the relapse setting daratumumab plus Rd has the longest progression-free survival (median 44.5 vs. 17.5 months) with the best hazard ratio (HR: 0.44), compared to other lenalidomide-based triplets such as Rd plus elotuzumab (median progression-free survival, 19.4 versus 14.9 months, HR 0.71), ixazomib (median progression-free survival, 20.6 months versus 14.7 months, HR: 0.74) or carfilzomib (median progression-free survival, 26.1 versus 16.6 months, HR: 0.66) [21–26]. Daratumumab and isatuximab can also effectively be combined with pomalidomide–dexamethasone in patients with prior lenalidomide exposure [27–29]. In the phase 3 APOLLO (median of two prior lines of therapy) and IKARIA (median of three prior lines of therapy) studies, there was roughly a doubling in response rate (≥PR: 46 versus 69% in APOLLO, and 35 versus 60% in IKARIA) and doubling in progression-free survival (APOLLO: median progression-free survival 6.9 versus 12.4 months, HR 0.63; IKARIA: median progression-free survival 6.5 versus 11.5 months, HR 0.60) when either daratumumab or isatuximab was added to the standard-of-care regimen pomalidomide–dexamethasone [28,29].

**Table 1.** Selected phase 3 studies evaluating IMiD-based antibody combinations.

| Study | Regimens | Patient Population |
|---|---|---|
| MAIA | Lenalidomide–dexamethasone +/− DARA | Newly diagnosed myeloma patients not eligible for immediate autologous stem cell transplant |
| POLLUX | Lenalidomide–dexamethasone +/− DARA | At least one prior line of therapy |
| APOLLO | Pomalidomide–dexamethasone +/− DARA | ≥1 prior line of therapy including lenalidomide and a proteasome inhibitor; patients with only 1 prior line of therapy were required to be refractory to lenalidomide. |
| ICARIA | Pomalidomide–dexamethasone +/− ISA | ≥2 previous lines of treatment, including lenalidomide and a proteasome inhibitor |
| ELOQUENT-2 | Lenalidomide–dexamethasone +/− ELO | 1–3 prior lines of therapy |
| ELOQUENT-3 | Pomalidomide–dexamethasone +/− ELO | ≥2 prior lines of therapy, including at least two consecutive cycles of lenalidomide and a proteasome inhibitor alone or in combination. |

Abbreviations: DARA, daratumumab; ISA, isatuximab; ELO, elotuzumab.

### 2.2.2. PI-Based Combinations

The proteasome inhibitors bortezomib and carfilzomib are effective partner drugs with CD38-targeting antibodies in patients with relapsed/refractory MM (Table 2). The

CASTOR study randomized patients with at least one prior line of therapy between bortezomib–dexamethasone and daratumumab plus bortezomib–dexamethasone [30]. The triplet regimen resulted in a higher response rate and superior progression-free survival across most subgroups [30]. The combination of carfilzomib–dexamethasone plus a CD38-targeting antibody was evaluated in two phase 3 trials in patients with 1–3 prior lines of therapy: isatuximab in the IKEMA and daratumumab in the CANDOR study [31,32]. In both studies, the triplet with a CD38-targeting antibody significantly improved response and progression-free survival with a favorable benefit-risk profile. In the IKEMA study, the overall response was 86% versus 83% with a median progression-free survival of not reached versus 19.2 months (HR: 0.53) in the treatment arm with isatuximab and without isatuximab, respectively. In the CANDOR study, the overall response rate was 84% versus 75%, and median progression-free survival was not reached versus 15.8 months (HR: 0.63) in the daratumumab arm and control arm, respectively. Adding a CD38 antibody to the carfilzomib–dexamethasone backbone also improved the proportion of patients who achieved minimal residual disease-negativity. Both triplets are excellent treatment options for patients with a first lenalidomide-refractory relapse [33].

**Table 2.** Selected phase 3 studies evaluating proteasome inhibitor-based antibody combinations.

| Study | Regimens | Patient Population |
|---|---|---|
| ALCYONE | Bortezomib–melphalan–prednisone (VMP) +/− DARA | Newly diagnosed myeloma patients not eligible for immediate autologous stem cell transplantation |
| CASTOR | Bortezomib–dexamethasone +/− DARA | At least one prior line of therapy |
| CANDOR | Carfilzomib–dexamethasone +/− DARA | 1–3 prior lines of therapy |
| IKEMA | Carfilzomib–dexamethasone +/− ISA | 1–3 prior lines of therapy |

Abbreviations: DARA, daratumumab; ISA, isatuximab.

### 2.2.3. CD38-Targeting Antibody Based Quadruplets

Younger transplant-eligible patients with newly diagnosed disease are frequently treated with high-dose melphalan and autologous stem cell transplantation. Prior to high-dose therapy, these patients receive induction therapy, typically a bortezomib-based triplet such as bortezomib–lenalidomide–dexamethasone (VRD), bortezomib–thalidomide–dexamethasone (VTD), or bortezomib–cyclophosphamide–dexamethasone (VCD) [34]. Because of the favorable activity and safety profile of CD38-targeting antibodies, several studies have evaluated or are currently evaluating the value of adding a CD38-targeting antibody to these triplets (Table 3). The CASSIOPEIA study showed the superiority of daratumumab plus VTD versus VTD alone before and as consolidation after transplantation [35]. The complete response (CR) rate was 39% in the daratumumab group and 26% in the control group, and 64% versus 44% a achieved minimal residual disease-negativity ($10^{-5}$ sensitivity threshold, assessed by multiparametric flow cytometry. The improved response rate resulted in a significantly improved progression-free survival (hazard ratio 0.47) [35]. The randomized phase 2 GRIFFIN study also showed a higher quality of response when daratumumab was added to VRD, compared to VRD alone [36]. In this study, there is not yet a progression-free survival advantage observed in the daratumumab-treated patients. The phase 3 PERSEUS study, which is ongoing, also compares VRD plus or minus daratumumab in a larger number of transplant-eligible patients. The combination of carfilzomib–lenalidomide–dexamethasone (KRd) plus a CD38-targeting antibody is also evaluated in transplant-eligible patients with newly diagnosed disease (e.g., in the ISKIA study). In addition, phase 3 trials (e.g., PERSEUS, CASSIOPEIA, and AURIGA) are also investigating CD38-targeting antibodies alone or in combination with lenalidomide as maintenance treatment post-transplant [35,36].

**Table 3.** Selected phase 3 studies evaluating IMiD + proteasome inhibitor-based antibody combinations.

| Study | Regimens | Patient Population |
|---|---|---|
| CASSIOPEIA | Bortezomib–thalidomide–dexamethasone +/− DARA | Newly diagnosed myeloma patients eligible for autologous stem cell transplant and aged ≤65 |
| PERSEUS | Bortezomib–lenalidomide–dexamethasone +/− DARA | Newly diagnosed myeloma patients eligible for autologous stem cell transplant and aged ≤70 |
| CEPHEUS | Bortezomib–lenalidomide–dexamethasone +/− DARA | Newly diagnosed myeloma patients for whom transplant is not intended as initial therapy |
| IMROZ | Bortezomib–lenalidomide–dexamethasone +/− ISA | Newly diagnosed myeloma patients not eligible for autologous stem cell transplantation |

Abbreviations: DARA, daratumumab; ISA, isatuximab.

Elderly, non-transplant eligible patients can be treated with several approved regimens such as the doublet lenalidomide–dexamethasone and the triplets daratumumab–lenalidomide–dexamethasone or bortezomib–melphalan–prednisone (VMP). In addition, based on results from the ALCYONE study, there is now one quadruplet regimen approved for these patients. In the ALCYONE study, non-transplant eligible MM patients received either nine cycles of bortezomib–melphalan–prednisone (VMP) alone or with daratumumab until progression [37,38]. The daratumumab group experienced superior clinical outcomes including an improved overall survival. Part of the improved outcomes with daratumumab–VMP may be related to the design of the study, with patients treated with VMP alone not receiving any maintenance after nine cycles of VMP, while daratumumab was continued until progression in the experimental arm. In addition, only a small proportion of patients who developed disease progression in the VMP only arm were treated with a CD38-targeting antibody at the time of relapse, which may explain part of the overall survival benefit in the daratumumab arm [37]. The combination of CD38-targeting antibody plus VRD is also evaluated in two phase 3 randomized trials with transplant-ineligible patients: CEPHEUS with daratumumab, and IMROZ with isatuximab. Importantly, a meta-analysis in newly diagnosed patients showed that daratumumab also improved progression-free survival in patients with high-risk cytogenetics such as del(17p), t(4;14), and t(14;16) [39].

### 2.2.4. Toxicity in Combination Regimens

When CD38-targeting antibodies are added to standard-of-care regimens, this is typically accompanied by a higher rate of infections. Particularly, the frequency of respiratory infections is increased. This may be caused by the reduction in already-suppressed polyclonal immunoglobulins [40] or NK cell depletion [41]. Furthermore, the higher rate of neutropenia, especially when a CD38-targeting antibody is combined with an IMiD, may contribute to the development of infections. Patients at high risk for infections (such as elderly patients, or those with elevated lactate dehydrogenase (LDH) low albumin, or elevated alanine aminotransferase (ALAT) may benefit from antibacterial prophylaxis [42].

### 2.2.5. Subcutaneous Administration

Daratumumab can also be administered via a 5 min subcutaneous injection [43]. The COLUMNA study demonstrated the non-inferiority of subcutaneous daratumumab versus intravenous daratumumab in terms of efficacy and pharmacokinetics [44]. Subcutaneous daratumumab also had an improved safety profile in patients with relapsed or refractory MM with a lower rate of infusion reactions [44]. Subcutaneous administration of daratumumab reduces the time spent in the outpatient infusion center, and thereby quality of life.

### 2.3. SLAMF7-Targeting Antibodies

Elotuzumab is the first-in-class naked SLAMF7-targeting antibody, which induces MM cell death via ADCC and ADCP [45]. Binding of elotuzumab to SLAMF7 present on the cell surface of NK cells results in NK cell activation and improved immune-mediated

attack of MM cells [46,47]. Elotuzumab has no single agent activity in heavily pretreated MM patients, but it significantly enhances the anti-MM effects of IMiDs and PIs in patients with relapsed/refractory MM [21,48–50].

There are currently two elotuzumab-based triplets approved for the treatment of relapsed/refractory MM patients: elotuzumab–lenalidomide–dexamethasone [21] and elotuzumab–pomalidomide–dexamethasone [48]. The ELOQUENT-2 study demonstrated a superior response rate, as well as longer progression-free and overall survival with elotuzumab added to lenalidomide–dexamethasone compared to lenalidomide–dexamethasone alone in patients with 1–3 prior lines of therapy [21,25,42,51]. Similarly, a higher response rate (53 versus 26%) and longer progression-free survival (median progression-free survival: 10.3 versus 4.7 months) was reported in the ELOQUENT-3 study for elotuzumab plus pomalidomide–dexamethasone, compared to pomalidomide–dexamethasone alone in patients with at least two prior lines of therapy including an IMiD and PI [48]. The toxicity profile of elotuzumab is mild, with a low rate of infusion reactions.

Surprisingly, until now, the incorporation of elotuzumab has not been successful in the setting of newly diagnosed disease. Adding elotuzumab to VRD in high-risk, newly diagnosed MM patients did not improve clinical outcomes [52]. In addition, the phase 3 ELOQUENT-1 study showed that elotuzumab plus lenalidomide–dexamethasone was not superior to lenalidomide–dexamethasone alone in newly diagnosed transplant ineligible patients. Finally, elotuzumab plus VRD as induction therapy prior to transplantation did not improve response rate or response quality in the large German GMMG-HD6 study [53].

## 3. Triple-Class Refractory Myeloma

Although the survival of MM patients significantly improved through the introduction of IMiDs, PIs, and CD38-targeting antibodies, virtually all patients eventually develop resistance towards these agents. MM patients with disease which is resistant to IMiDs, PIs, and CD38-targeting antibodies (triple-class refractory disease) have a very poor outcome at this moment. One analysis showed that triple-class refractory patients have a median overall survival of less than 12 months [54]. Penta-refractory patients (disease refractory to lenalidomide, pomalidomide, bortezomib, carfilzomib, and a CD38-targeting antibody) have the worst prognosis, with a median overall survival of only 5.6 months [54].

These triple-class refractory patients benefit from newly approved drugs with novel mechanism of action such as selinexor, which inhibits XPO-1-mediated nuclear export [55], or the immunoconjugate belantamab mafodotin (see next section) [56]. One study with 34 patients (median of three prior lines of therapy) showed that selinexor can also be effectively combined with daratumumab and dexamethasone [57]. Common adverse events with this IMiD- and PI-free regimen included thrombocytopenia, nausea, and fatigue. The overall response rate was 73% with a median progression-free survival of 12.5 months in daratumumab-naïve patients. In addition, retreatment with drugs that were used in prior lines of therapy can also be considered [33]. However, these triple-class refractory patients should also be considered for clinical trial participation. In such trials, several promising new immunotherapies are evaluated, including chimeric antigen receptor (CAR) T cells and cereblon E3 ligase modulators (CelMods), but also with new antibody formats such as immunoconjugates and bispecific antibodies. The most common target for immunoconjugates and bispecific antibodies is B cell maturation antigen (BCMA). This cell surface protein is highly and uniformly expressed on normal plasma cells, MM cells, and a small subset of mature B cells. BCMA promotes MM cell survival and proliferation [58]. The first naked BCMA antibody evaluated in preclinical studies was SG1 [59]. Although it effectively eliminated MM cells, it was not further developed. A phase 1 study evaluating a humanized, non-fucosylated IgG1 anti-BCMA naked antibody is ongoing [60]. However, maybe more important is that the rather selective expression of BCMA on MM cells makes it possible to improve the cytotoxic capacity of the antibody. Indeed, at this moment powerful immunotherapeutic drugs targeting BCMA show promising results in heavily pretreated patients.

## 4. Immunoconjugates

Antibodies can also be used to specifically deliver a small molecule (antibody–drug conjugate (ADC)), toxin (immunotoxin), cytokine (immunocytokine) or radionuclide (radioimmunoconjugate) to the tumor cells (Figure 1) [61]. Several immunoconjugates are being investigated in preclinical or phase 1 clinical studies [61], but most advanced in its development is the ADC belantamab mafodotin.

Belantamab mafodotin is a BCMA-directed antibody conjugated by a protease-resistant maleimidocaproyl linker to the microtubule-disrupting agent, monomethyl auristatin F (MMAF). BCMA is specifically expressed on normal plasma cells, MM cells, and a small subset of mature B-cells; therefore, MMAF will be selectively targeted to the tumor cells. Belantamab mafodotin also has other modes of action, such as ADCC and the inhibition of BCMA receptor signaling. In the DREAMM-2 study, belantamab mafodotin was administered every three weeks to triple-class refractory patients. In the 2.5 mg/kg cohort the overall response rate was 31% with a median progression-free survival of 2.9 months, while in the 3.4 mg/kg cohort this was 34% and 4.9 months [56]. The approved dose for use in patients with at least four prior lines of therapy, including a CD38-targeting antibody, PI and IMiD, is 2.5 mg/kg, based on the lower rate of adverse events leading to dose delay or dose reductions in the 2.5 mg/kg cohort, compared to the 3.4 mg/kg cohort [56]. Toxicity of belantamab mafodotin consists of thrombocytopenia and corneal toxicity (keratopathy). Because of the corneal adverse events, close collaboration with an ophthalmologist is important, with ocular assessments at baseline and every cycle thereafter. Based on the activity observed in advanced MM, several combination studies with belantamab mafodotin are ongoing, also in earlier stages of the disease. This includes the DREAMM-6 study, which evaluates the combination of belantamab mafodotin with lenalidomide–dexamethasone or bortezomib–dexamethasone in patients with at least one prior line of therapy [62]. Preliminary data show an acceptable safety profile of belantamab mafodotin (2.5 mg/kg) plus bortezomib–dexamethasone, with keratopathy and thrombocytopenia as the most common adverse events. The three-drug combination induced an overall response rate of 78% in these patients with a median of three prior lines of therapy. Based on these results, the phase 3 DREAMM-7 study is now enrolling patients to evaluate bortezomib–dexamethasone with or without belantamab mafodotin in patients with at least one prior therapy. DREAMM-8 is another phase 3 study, which randomizes patients with at least one prior line of therapy (including lenalidomide) to pomalidomide–dexamethasone or pomalidomide–dexamethasone with belantamab mafodotin. Newly diagnosed transplant-ineligible MM patients can be enrolled in the DREAMM-9 study, which compares VRD with or without belantamab mafodotin.

Several other BCMA-targeting ADCs are in development, including AMG-224 and MEDI2228 [63,64]. AMG-224 is a BCMA antibody conjugated to the tubulin inhibitor mertansine (DM1). In a phase 1 study with 42 patients with a median of 7 prior lines of therapy, the overall response rate was 23% and the recommended phase 2 dose was determined as 3 mg/kg [63]. Ocular adverse events were reported in 21% in the escalation cohort and 36% in the expansion cohort, but no dose reduction or delays were reported due to ocular events [63]. MEDI2228 is another BCMA-specific ADC with a DNA cross-linking pyrrolobenzodiazepine (PBD) dimer as a warhead, which is currently under clinical evaluation [64]. The overall response rate in 41 patients (56% triple-class refractory) treated at the maximum tolerated dose (0.14 mg/kg, every three weeks) was 66% with a median duration of response of 5.9 months. Although keratopathy was not reported in the 0.14 mg/kg cohort, photophobia was commonly observed (all grade: 59%; grade $\geq$ 3: 17%). ADCs targeting other MM-associated antigens are also in (pre)clinical development, such as ADCs targeting CD38, CD138, CD46, and FcRL5 [61].

Importantly, several immunotoxins and immunocytokines are also in early phases of clinical development. This includes TAK-169, which comprises an anti-CD38 single-chain variable fragment fused to the Shiga-like toxin A-subunit [65], and TAK-573, a CD38-specific IgG4 antibody fused to an attenuated form of human IFNα2b [66]. TAK-573

not only has direct anti-tumor activity, but also has the ability to enhance immune cell function [67].

## 5. Bispecific Antibodies

Durable remissions in a subset of MM patients, who received an allogeneic stem cell transplantation (allo-SCT) or a donor lymphocyte infusion (DLI), provided evidence of the existence of a graft-versus-myeloma effect mediated by donor T cells [68–71]. However, because allo-SCT and DLI are also associated with the development of sometimes life-threatening, infections and graft-versus-host disease, several new strategies that use T cells to eliminate MM cells have been developed with high potency and a better safety profile, compared to allo-SCT. This includes T cells genetically modified to express a chimeric antigen receptor (CAR) that targets a surface antigen expressed on the MM cell. BCMA-specific CAR T cells are very promising, with high response rates and durable responses observed in heavily, often triple-class refractory, MM patients [72–76]. An alternative, off-the-shelf approach to redirect T cells to MM cells is the application of bispecific antibodies (Figure 1) [77,78].

The first-in-class T cell redirecting antibody used in MM is AMG-420 (Table 4) [79]. AMG-420 is a bispecific T cell engager, comprising two single-chain fragment variables and a peptide linker, and lacking an Fc domain. AMG-420 binds with one arm to the CD3 antigen present on the T cell surface, and with the other arm to BCMA, present on the MM cell. This coupling of T cells and MM cells results in T cell activation and degranulation, and subsequently, MM cell death. AMG-420 has a short half-life and needs to be administered via continuous intravenous infusion over four weeks of each six-week cycle [79]. The overall response rate of AMG-420 in patients (median of 3.5 prior lines of therapy), treated at the maximum tolerated dose was 70%, with cytokine release syndrome observed in 38%. Other side effects included infections, cytopenias, and polyneuropathy [79]. Clinical development of AMG-420 stopped because of the need for continuous infusion, which can be challenging for patients. A half-life extended variant, AMG-701, which can be administered once weekly, is now being evaluated in clinical trials with promising results from the phase 1 trial. The overall response rate was 83% in the most recent evaluable cohort, with four out of five responders being triple-class refractory [80].

Teclistamab is another IgG-like bispecific antibody with high activity in advanced MM [81]. In an ongoing phase 1 study, it is administered via intravenous (I.V.) infusion or subcutaneous (S.C.) injection [82]. Most active doses were 270–720 µg/kg I.V. and 720–3000 µg/kg S.C., with an overall response rate at these doses of 69% (overall response rate: 67% (18/27) in I.V. cohorts and 71% (29/41) in S.C. cohorts). The overall response rate at the recommended phase 2 dose of 1500 µg/kg (S.C. administration) was 73%, including at least a very good partial response (VGPR) in 55% of the patients ($n = 22$, 85% triple-class refractory) [82]. These responses were durable and improved over time. Teclistamab was well-tolerated at the dose of 1500 µg/kg S.C., with only grade 1 or 2 cytokine release syndrome (CRS) events, mainly occurring following the step-up dosing or first full dose. A phase 2 expansion study has started based on these promising results.

Preliminary results from other studies also show promising activity of IgG-like BCMA-targeting bispecific antibodies such as CC-93269, which is characterized by bivalent binding to BCMA [83]. Response to CC-93269 was dose-dependent, with an overall response rate of 43% in all patients ($n = 30$; 67% triple-class refractory) and 89% in the nine patients treated with 10 mg CC-93269, including CR in 44%. CRS occurred in 77% of patients, including one grade $\geq 3$ event [83]. This study is ongoing to define the recommended phase 2 dose. Other BCMA/CD3-bispecific antibodies in clinical development are PF-06863135, REGN5458, and TNB-383B (see Table 4) [84–86].

**Table 4.** Selected studies with bispecific T cell engagers.

| Drug Name | Company | Target | Format | Phase of Study | Administration Route | Number of Patients | Median Age (years) | Triple Class Refractory (%) | CRS (All Grade) (%) | CRS (Grade ≥ 3) (%) | ≥PR | ≥VGPR |
|---|---|---|---|---|---|---|---|---|---|---|---|---|
| AMG-420 [79] | AMGEN | BCMA | BiTE | 1 | Continuous I.V. infusion | 42 | 65 | ≤21 | 38 | 2 | 70% at the MTD of 400 ug/day (n = 10) | 60% at the MTD of 400 ug/day (n = 10) |
| AMG-701 [80] | AMGEN | BCMA | Half-life extended BiTE | 1 | I.V. | 85 | 64 | 62 | 65 | 9 | 83% in most recent evaluable cohort (n = 6) | 50% in most recent evaluable cohort (n = 6) |
| Teclistamab [82] | Janssen Pharmaceuticals | BCMA | Bispecific antibody | 1 | I.V. or S.C. | 149 | 63 | 81 | 55 | 0 | 73% at the RP2D (1500 µg/kg SC) (n = 22) | 55% at the RP2D (1500 µg/kg SC) (n = 22) |
| CC-93269 [83] | BMS/Celgene | BCMA | Bispecific antibody | 1 | I.V. | 30 | 64 | 67 | 77 | 3 | 89% among patients with 10 mg (n = 9) | 78% among patients with 10 mg (n = 9) |
| REGN5458 [85] | Regeneron | BCMA | Bispecific antibody | 1 | I.V. | 49 | 64 | 100 | 39 | 0 | 63% at dose level 6 (n = 8) | 63% at dose level 6 (n = 8) |
| PF-06863135 [84] | Pfizer | BCMA | Bispecific antibody | 1 | I.V. and S.C. | 30 | 63 | NR | 73 | 0 | 80% at the 215–1000 µg/kg SC dose (n = 20) | NR |
| TNB-383B [86] | Tenebio | BCMA | Bispecific antibody | 1 | I.V. | 58 | 66 | 64 | 45 | 0 | 80% at dose of 40–60 mg (n = 15) | 73% at dose of 40–60 mg (n = 15) |
| Talquetamab [87] | Janssen Pharmaceuticals | GPRC5D | Bispecific antibody | 1 | I.V. or S.C. | 157 | 64 | 82 | 54 | 3 | 69% at the RP2D (405 µg/kg SC) (n = 13) | 39% at the RP2D (405 µg/kg SC) (n = 13) |
| Cevostamab [88] | Roche/Genentech | FcRH5 | Bispecific antibody | 1 | I.V. | 53 | 62 | 72 | 78 | 2 | 53% in ≥3.6/20 mg cohorts (n = 34) | 32% in ≥3.6/20 mg cohorts (n = 34) |

Abbreviations: PR, partial response; VGPR, very good partial response; S.C., subcutaneous; I.V., intravenous; RP2D, recommended phase 2 dose; MTD maximum-tolerated dose; NR, not reported.

Several studies are also evaluating bispecific antibodies targeting other MM-associated antigens such as GPRC5D and FcRH5. Talquetamab is the first-in-class GPRC5D-targeting bispecific antibody with high activity in triple-class refractory MM [87]. In an ongoing phase 1 study, talquetamab is administered once weekly via intravenous infusion or subcutaneous injection. Talquetamab has a tolerable safety profile at the recommended phase 2 dose of 405 μg/kg S.C. Frequent adverse events include CRS (no grade 3 events reported with S.C. dosing) and skin toxicity including nail disorders. The overall response rate in the 19 patients (68% triple-class refractory) treated with 405 μg/kg S.C. was 69%, including at least VGPR in 39% [87]. Finally, cevostamab is the first-in-class FcRH5-targeting bispecific antibody which is administered intravenously every three weeks [88]. Preliminary results of the first 53 patients (72% triple-class refractory) have been reported [88]. The overall response rate was 53% in 34 patients who received active doses. CRS was observed in 76% of patients (grade $\geq$ 3 in 2%).

## 6. Conclusions

The last decade has demonstrated substantial progress in immunotherapy of MM patients. Firstly, the incorporation of CD38-targeting antibodies into standard-of-care relapse and frontline regimens has markedly improved the outcomes of MM patients. In addition, more recently, several clinical studies have shown that new antibody formats such as immunoconjugates and bispecific antibodies have high activity in extensively pretreated, often triple-class refractory, patients. Given the high activity of these new immunotherapies, several ongoing studies are evaluating the value of these novel therapies in earlier phases of the disease (early relapse as well as newly diagnosed disease), frequently combined with other anti-MM agents, such as CD38 antibodies or IMiDs. Given the high single agent activity of bispecific antibodies, these agents may also be applied in patients, who remain minimal residual disease-positive after optimal induction therapy with or without transplant. Efforts are also ongoing to mitigate eye toxicity associated with the BCMA-targeting ADCs. Another open research question is which patients will benefit most from BCMA-targeting CAR T cells, bispecific antibodies, or ADCs. In this respect, not only are differences in efficacy of importance, but safety aspects also play a role. Pre-existing ocular toxicity may limit the applicability of belantamab mafodotin, while compromised cardio-pulmonary function may limit the use of bispecific antibodies or CAR T cells, which often induce CRS. Another important aspect is the direct "off-the-shelf" availability of bispecific antibodies and ADCs, while CAR T cell therapy needs more time for manufacturing. Patients with rapidly progressing disease may benefit most from such "off-the-shelf" approaches. However, in the future, allogeneic CAR T cells or NK cells may be able to overcome such logistical issues. The role of immune fitness also deserves further investigation, because the use of T cell redirection therapy with bispecific antibodies or CAR T cells early in the disease course may be more effective than in end-stage MM, where the cumulative exposure to immunosuppressive anti-MM agents has resulted in substantial impairment of T cell function. In the near future, we will learn whether earlier application of these novel immunotherapies, in combination with other agents, will lead to further improvements in the survival of MM patients.

**Author Contributions:** C.P.M.V., W.S.C.B., S.Z. and N.W.C.J.v.d.D. equally contributed to writing the manuscript and approved the final version. All authors have read and agreed to the published version of the manuscript.

**Funding:** This research received no external funding.

**Institutional Review Board Statement:** Not applicable.

**Informed Consent Statement:** Not applicable.

**Conflicts of Interest:** N.W.C.J.v.d.D.: Research support from Janssen Pharmaceuticals, Amgen, Celgene, Novartis, Cellectis, and Bristol-Myers Squibb; Advisory boards for Janssen Pharmaceuticals, Amgen, Celgene, Bristol-Myers Squibb, Novartis, Roche, Takeda, GSK, Sanofi, Bayer and Servier.

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
