# Peer review of "Immunotherapy with Antibodies in Multiple Myeloma: Monoclonals, Bispecifics, and Immunoconjugates"

_hemato, doi:10.3390/hemato2010007_

Round 1

Reviewer 1 Report

The present review article by Drs Verkleij et al is well written and up-to-date. I have only few comments in order to further improve the readability of the manuscript.

-The abstract should be modified. Authors should state which immunotherapies (naked Abs and ADCs) are FDA/EMA approved. At the moment the focus lies very much on CD38- directed antibodies, not on monoclonals, bispecifics, and immunoconjugates.

-Line 35: Why did the authors include TAK-079 here? It is not yet approved; other CD38 Abs such as MOR202, SAR 442085, TAK-573, … are not mentioned. Would highlight those CD38-targeting Abs that are approved; and those which have been investigated and are in the clinical pipeline.

-3. Triple-class refractory myeloma:

Ongoing efforts to combine selinexor with mAbs could be discussed.

-Data on the ICARIA and APOLLO trial should be discussed in more detail.

-Line 180: since specific names of key studies are mentioned throughout the manuscript include: “ELOQUENT-3”

-To increase the readability of this manuscript include a table listing:

1) Ab- containing Rd combinations

2) Ab- containing Pd combinations

3) Ab- containing proteasome inhibitor combinations

both in ND and RR MM.

  • n, median age, prior lines of therapy, IMiD/ PI- refractory/ exposed, MRD(-), state of development

-shortly interpret data of the CANDOR versus CASTOR trial

-Include a table listing preliminary key findings on studies evaluating bispecific antibodies

  • n, response rates, prior lines of therapy, MRD(-)

-Include data on REGN5458, TNB-383B, and BFCR4350A which were presented at last year’s ASH meeting

-Belantamab mafodotin: Preliminary data on DREAMM-1 to 6 should be mentioned

-Line 275: AMG-720?

-For completeness: authors should also mention BCMA/CD3- bispecific antibodies PF-06863135, REGN5458, and TNB-383B

-Shortly discuss potential future strategies involving immunotherapies (BiTes, naked Abs, ADCs) and CAR T cells.

Author Response

The present review article by Drs Verkleij et al is well written and up-to-date. I have only few comments in order to further improve the readability of the manuscript.

-The abstract should be modified. Authors should state which immunotherapies (naked Abs and ADCs) are FDA/EMA approved. At the moment the focus lies very much on CD38- directed antibodies, not on monoclonals, bispecifics, and immunoconjugates. REPLY: We thank the reviewer for carefully reviewing our manuscript. We have added in the abstract that CD38 antibodies and elotuzumumab as well as belantamab mafodotin are approved treatment options in MM.

-Line 35: Why did the authors include TAK-079 here? It is not yet approved; other CD38 Abs such as MOR202, SAR 442085, TAK-573, … are not mentioned. Would highlight those CD38-targeting Abs that are approved; and those which have been investigated and are in the clinical pipeline. REPLY: this section discusses the naked CD38 antibodies. We have as suggested indicated that dara and isa are approved, while TAK-079, SAR442085 and MOR202 are not approved at this moment but in clinical development. TAK-573 is discussed in the section on immunoconjugates.

-3. Triple-class refractory myeloma:

Ongoing efforts to combine selinexor with mAbs could be discussed.

REPLY: We have added data showing that selinexor can be effectively combined with CD38 antibodies (see page 7).

-Data on the ICARIA and APOLLO trial should be discussed in more detail. REPLY: we have added more data on ICARIA and APOLLO, including response rate and median PFS.

-Line 180: since specific names of key studies are mentioned throughout the manuscript include: “ELOQUENT-3” REPLY: As suggested, we have added ELOQUENT-3 here.

-To increase the readability of this manuscript include a table listing:

1) Ab- containing Rd combinations

2) Ab- containing Pd combinations

3) Ab- containing proteasome inhibitor combinations

both in ND and RR MM.

REPLY: we have added 3 tables in which we mention the important phase 3 studies evaluating IMID-based antibody combinations, PI-based antibody combinations, and IMiD/PI-based antibody combinations. We have included the key characteristics of the patient population that was studied.

  • n, median age, prior lines of therapy, IMiD/ PI- refractory/ exposed, MRD(-), state of development

-shortly interpret data of the CANDOR versus CASTOR trial REPLY: we have added more data on CANDOR and IKEMA in the corresponding section.

-Include a table listing preliminary key findings on studies evaluating bispecific antibodies REPLY: we have include a table with data on bispecific T-cell engagers (Table 4).

  • n, response rates, prior lines of therapy, MRD(-)

-Include data on REGN5458, TNB-383B, and BFCR4350A which were presented at last year’s ASH meeting  REPLY: we included these compounds in the manuscript text as well in the new table 4.

-Belantamab mafodotin: Preliminary data on DREAMM-1 to 6 should be mentioned  REPLY: We have added information on some of the other DREAMM studies, such as DREAMM-6, 7, and 8.

-Line 275: AMG-720?  REPLY: we have changed this in AMG-701.

-For completeness: authors should also mention BCMA/CD3- bispecific antibodies PF-06863135, REGN5458, and TNB-383B REPLY: we added these compounds in the manuscript text as well in the new table 4 (with data on bispecific T-cell engagers).

-Shortly discuss potential future strategies involving immunotherapies (BiTes, naked Abs, ADCs) and CAR T cells.  REPLY: On page 12, in the conclusion section, we have included a brief discussion on future strategies with ADCs, bispecifics.

Reviewer 2 Report

This is a well-written summary of the currently approved antibody-based therapies as well as the newer bispecific agents under development for myeloma.

Minor points:

  1. Page 3, lines 90-92. Comparison of agents across different clinical trials cannot be used to determine superiority of an agent/combination. Rephrase the sentence regarding Dara-Rd to eliminate the judgement regarding potency and instead provide the median PFS and HR values for DaraRd, Elo-Rd, KRd and IRd.
  2. Page 4, discussion of ALYCONE. Authors should provide further detail regarding the study design, which likely was a major factor in the outcomes, namely that patients in the VMP arm received fixed duration therapy while patients in the Dara-VMP arm went on to receive single agent daratumumab until PD.
  3. Line 163. Suspect that this is lost-in-translation, but in the US, “day-care unit” means something entirely different.
  4. Before jumping into discussion of the BCMA ADC/bispecifics, the authors should briefly discuss BCMA as a target and the failure of the naked antibody approach, thus necessitating the ADC or bispecific approach.

Author Response

This is a well-written summary of the currently approved antibody-based therapies as well as the newer bispecific agents under development for myeloma.

Minor points:

  1. Page 3, lines 90-92. Comparison of agents across different clinical trials cannot be used to determine superiority of an agent/combination. Rephrase the sentence regarding Dara-Rd to eliminate the judgement regarding potency and instead provide the median PFS and HR values for DaraRd, Elo-Rd, KRd and IRd.  REPLY: We thank the reviewer for carefully reviewing our manuscript. We rephrased this sentence and added median PFS and HR for all 4 studies evaluating lenalidomide-dexamethasone plus a third drug.
  2. Page 4, discussion of ALYCONE. Authors should provide further detail regarding the study design, which likely was a major factor in the outcomes, namely that patients in the VMP arm received fixed duration therapy while patients in the Dara-VMP arm went on to receive single agent daratumumab until PD. REPLY: WE agree with the reviewer that designs plays an important role in the improved outcomes in the dara-VMP arm. We therefore added: “Part of the improved outcomes with daratumumab-VMP may be related to the design of the study, with patients treated with VMP alone not receiving any maintenance after 9 cycles of VMP, while daratumumab was continued until progression in the experimental arm.”.
  3. Line 163. Suspect that this is lost-in-translation, but in the US, “day-care unit” means something entirely different. REPLY: We changed day care unit into outpatient infusion center. I think (and hope) this is a more commonly used term, also in USA.
  4. Before jumping into discussion of the BCMA ADC/bispecifics, the authors should briefly discuss BCMA as a target and the failure of the naked antibody approach, thus necessitating the ADC or bispecific approach. REPLY: We have added extra information on BCMA (expression and function), and also describe that the selective expression of BCMA allows for enhancing the cytotoxic capacity of the antibody. We also describe that clinical development of the first naked antibody targeting BCMA was stopped.

Reviewer 3 Report

The authors summarized the Immunotherapy strategies with antibodies in multiple myeloma. The paper is interesting and well written. 

I suggest to add a paragraph or table summarizing the pros and cons of the immunotherapy strategies described in this review. 

Author Response

Comments and Suggestions for Authors

The authors summarized the Immunotherapy strategies with antibodies in multiple myeloma. The paper is interesting and well written. 

I suggest to add a paragraph or table summarizing the pros and cons of the immunotherapy strategies described in this review. 

We thank the reviewer for carefully reviewing our manuscript. We have added in the conclusion section a paragraph on pros and cons of the different immunotherapeutic strategies.